Effects of astragalus polysaccharide on the adhesion-related immune response of endothelial cells stimulated with CSFV in vitro

Zhuge Zengyu 1
Dong Yanpeng 2
Li Liuan 1
Jin Tianming jtmsci@163.com 1
1 Animal Science and Veterinary Medicine College, Tianjin Agricultural University , Tianjin , China
2 College of Forestry, Henan Agricultural University , Zhengzhou , Henan , China
Albertini Maria Cristina
Electronic publication date: 2017 Oct 6
Publication date: 2017
Volume: 5
Electronic Location ID: e3862
Received 2017 Jun 10; Accepted 2017 Sep 7
Copyright: ©2017 Zhuge et al.
Copyright year: 2017
Copyright holder: Zhuge et al.
License: This is an open access article distributed under the terms of the Creative Commons Attribution License, which permits unrestricted use, distribution, reproduction and adaptation in any medium and for any purpose provided that it is properly attributed. For attribution, the original author(s), title, publication source (PeerJ) and either DOI or URL of the article must be cited.
License URL: https://creativecommons.org/licenses/by/4.0/

Keywords: CSFV, Astragalus polysaccharide, Endothelial cell, E-selectin, P-selectin

Funding: Tianjin Municipal Education Commission, China 20120621 Veterinary Biotechnology Scientific Research Innovation Team of Tianjin, China TD12-5019 National Natural Science Foundation of China 31572492 31072109 This work was supported in part by the Fund for Development Projects of Science and Technology in High school from Tianjin Municipal Education Commission, China (grant number 20120621); the Veterinary Biotechnology Scientific Research Innovation Team of Tianjin, China (grant number TD12-5019) and National Natural Science Foundation of China (31572492, 31072109). There was no additional external funding received for this study. The funders had no role in study design, data collection and analysis, decision to publish, or preparation of the manuscript.

==============================
Background

Astragalus polysaccharide (APS) has immunomodulatory activities on porcine peripheral blood mononuclear cells. The immunomodulatory effects of APS on porcine endothelial cells (ECs) expose to classical swine fever virus (CSFV) remain unknown.

Methods

The virus was titrated using an indirect immune biotin enzyme standard method to confirm that porcine ECs were susceptible to CSFV infection and to determine the TCID50 of CSFV (C-strain). Porcine ECs were cultured with CSFV in the presence of APS. Relative quantitative PCR was used to assess the mRNA expression of factors that influence EC adhesion and immunity.

Results

The expression of adhesion factors mRNA increased following stimulation with CSFV; this effect was inhibited by pre-exposing the cells to APS. In addition, the expression of growth factors and some immune factors increased after infection with CSFV; this increase in tissue factor (TF), transforming growth factor (TGF-β), and interleukin-8 (IL-8) could be inhibited by the addition of APS. The immune response mediated by Toll-like receptor 4 (TLR4) in ECs may be unregulated by CSFV as it was also inhibited by pre-treatment with APS.

Discussion

The addition of APS to the culture can obviously regulate the expression of molecules related to the adhesion, growth, and immune response of ECs, as well as the production of cytokines. Therefore, it may have the potential to be an effective component in vaccines against CSFV.

Introduction

Classical swine fever (CSF) is a highly contagious disease affecting swine and leads to tremendous economic losses worldwide. The virus of CSF (CSFV) is a member of the Pestiviru s genus of the Flaviviridae family, which also includes viruses that cause bovine viral diarrhea/mucosal disease and border disease (Paton & Greiser-Wilke, 2003).

A CSFV infection in pigs causes hemorrhagic disease characterized by disseminated intravascular coagulation, thrombocytopenia, and immunosuppression (Penrith, Vosloo & Mather, 2011). Many different cell types have been studied in investigations of CSFV including macrophages (Dong et al., 2013; Borca et al., 2008). Vascular endothelial cells (ECs) maintain a hemostatic balance by providing a quiescent, anti-thrombotic barrier (Sasaki & Toyoda, 2013). ECs can be infected by CSFV and finally induces vascular diffuse lesions in the host (Ning et al., 2016; He et al., 2014). However, ECs can be rapidly activated by pathogens to express a proinflammatory and procoagulant phenotype to eliminate infection (Bierhaus & Nawroth, 2003). Evidence supports the notion that ECs play a critical role in the recruitment of immune cells, as well as influencing the outcome of the immune response (Rosemblatt & Bono, 2004; Pasyk & Jakobczak, 2004; Funayama et al., 1998).

Some factors can modulate EC function, including vascular endothelial growth factor (VEGF), VEGF is a multifunctional cytokine that plays a critical role in the regulation of angiogenesis and is also a potent mitogen that specifically targets ECs (Paton & Greiser-Wilke, 2003). The activation of tissue factor (TF) may regulate the initiation of the coagulation protease cascade in various cellular settings (Penrith, Vosloo & Mather, 2011) and is essential for triggering angiogenesis via VEGF production by human fibroblasts (Bierhaus & Nawroth, 2003).

Cell adhesion molecules (CAMs) including P-selectin and E-selectin, which are involved in cell tropism and pathogenesis during the course of viral infections (Nishimura et al., 2009). The expression of tumor necrosis factor (TNF)-α may contribute to the leukopenia in pigs infected with CSFV by promoting apoptosis (Rosemblatt & Bono, 2004). CSFV induces proinflammatory cytokines and tissue factor (TF) expression and inhibits apoptosis and interferon synthesis during the infection of porcine vascular ECs: The virus infection inhibited polyinosinic-polycytidylic acid [poly(I:C)]-induced α/β IFN (type I IFN) synthesis and there was a 100-fold increased capacity to resist apoptosis induced by poly(I:C) when the porcine kidney cell line SK-6 was infected with CSFV (Pasyk & Jakobczak, 2004). Type I IFNs have a bifunctional role in limiting the spread of the virus and inducing an antiviral state in uninfected cells while promoting apoptosis in infected cells. Furthermore, virus-induced apoptosis can be inhibited by anti-IFN-α/β antibodies (Ferrara & Davis-Smyth, 1997).

Endothelial-to-mesenchymal transition (EMT) contributes to fibrosis, which is a major cause of human death and disability (Ferrara & Keyt, 1997; Dvorak et al., 1995). The progression of fibrosis due to aortic banding is associated with the increased expression of mRNA encoding transforming growth factor-β1 (TGF-β1) (Drake & Ruf, 1989).

Astragalus polysaccharide (APS), which have been identified as an acidic heteropolysaccharide mainly composed of glucose, galactose, arabinose, rhamnose and small amount of mannose, fructose, ribose, xylose, fucose, glucuronic acid, and galacturonic acid, is extracted from Astragalus membranaceus (AM) roots and generally purified by dissolving in the distilled water, dialysis and lyophilization (Chen et al., 2015). APS is the main active ingredient in AM with molecular weights in the range of 8.7 × 103–4.8 × 107 g mol−1, Lai et al. (2017) has a substantial effect on exerting immunomodulatory effects in macrophage cells (Zhou et al., 2017), activating the immune system by clearing immune complexes (Jiang et al., 2010), enhancing the transformation of T lymphocytes, as well as activating B lymphocytes and dendritic cells (DC) (Shao et al., 2006; Liu et al., 2011). Moreover, APS has been used as an immunomodulator for the vaccination of pigs and to enhance the protective efficacy against foot-and-mouth disease virus (FMDV) (Li et al., 2011). Orally administered APS has also been shown to significantly enhance the efficacy of FMDV vaccination and has important implications for the further use of APS as a novel adjuvant (Zhang et al., 2010). Moreover, it was reported that AM markedly attenuated the inhibition of vasorelaxation and downregulation of cGMP levels in ECs, and APS exhibited a tendency to reverse both depressive responses (Zhang et al., 2007). Furthermore, APS plays a role in attenuating the infiltration of inflammatory cells (e.g., neutrophils and monocytes) by decreasing the expression of adhesion molecules, the release of chemokines, and induction of enzymes (Li, Zhang & Zhao, 2007). Some of the genes altered by APS treatment are associated with the activation of the immune proinflammatory response, such as the promotion of interleukin-1, 8, and interferon-γ production. Thus, APS serves to improve immune defense functions while resisting the invasion of external pathogens (Wei et al., 2011).

In our previous study, we found that APS exhibited immunomodulatory effects on cells exposed to CSFV and had the potential to be an effective adjuvant in vaccines against CSFV (Zhuge et al., 2012). In the present study, we evaluated the effect of APS on the cellular responses following a CSFV with the aim of identifying a potential method of combating this hemorrhagic infection.

Materials and Methods

Astragalus polysaccharides and virus

APS (freeze-dried) was kindly provided by Dr. Fenghua Liu (Beijing University of Agriculture, Beijing, China) and stored at room temperature. A rabbit propagated C-strain of CSFV was kindly provided by Dr. Hongwu Lang (China Institute of Veterinary Drug Control, Beijing, China) and stored at −20 °C.

Virus titration by an indirect immune biotin enzyme standard method

Porcine hip artery endothelial cells (ECs) were purchased from the China Institute of Veterinary Drug Control in Beijing. ECs (1 ×105/mL) were incubated in M199 culture medium (Invitrogen, Carlsbad, CA, USA) containing 2 mM L-glutamine enriched with 100 U/ml penicillin/streptomycin (Sigma, St. Louis, MO, USA) and 10% heat-inactivated fetal calf serum. These were incubated in a 96-well microplate (160 μL/well) for 24 h (5% CO2, 37 °C). The CSFV stock was diluted 10-fold × 10 (from 5 mg/mL to 5 ×10−10 mg/mL), and added to the microplate at 40 µL/well, with each concentration occupying eight wells. After 48 h incubation (5% CO2, 37 °C), the culture medium was aspirated and the plate was washed with PBS.

Hyperimmune rabbit-anti-CSFV serum (China Institute of Veterinary Drug Control, Beijing) was inactivated at 56 °C for 30 min before adding it to the culture plate (1:400) at 37 °C. The plates were washed with PBS three times after 1 h. Biotin-labeled sheep anti-swine antibodies (China Institute of Veterinary Drug Control, Beijing) were added to the culture plate (20 μg/mL, 100 μL/well) at 37 °C. Horseradish peroxidase-labeled avidin-biotin-peroxide complex (ABC) (5mg/mL) was added (100 μL/well) after 1 h. The culture plates were washed with PBS three times after 30 min. 3,3N-Diaminobenzidine Tetrahydrochloride (China Institute of Veterinary Drug Control, Beijing) was added (50 μL/well) to develop the color.

Cell cultures

ECs were cultured in M199 culture medium in 25 cm2 culture dishes. A final concentration of 103 TCID50/mL CSFV or 10 μg/mL APS was added after 48 h subculturing. Cells were allocated to one of six groups (each group had three replicates): (1) Medium (cells cultured only in medium); (2) APS pre-treatment (APS added 24 h after subculturing); (3) APS (APS added 48 h after subculturing); (4) CSFV (CSFV added 48 h after subculturing); (5) CSFV + APS pre-treatment (APS added 24 h and CSFV added 48 h after subculturing); (6) CSFV + APS (CSFV and APS both added 48 h after subculturing). The cells and supernatants in each of the six groups were harvested at 3 h and 6 h after the addition of CSFV (48 h after subculturing).

RNA isolation and reverse transcription

Trizol reagent (Invitrogen, Carlsbad, CA, USA) was used to extract the total RNA from the ECs, and the extracted RNA was dissolved in RNase-free water (Qiagen, Valencia, CA, USA). The integrity of each RNA sample was ascertained by agarose gel electrophoresis. The RNA purity (OD260/OD280 absorption ratio > 1.9) and its quantity were determined and verified using a NanoDrop® ND-2000C Spectrophotometer (NanoDrop Technologies Inc., Wilmington, DE, USA).

The reverse transcription (RT) reaction was initiated by adding 2 μg of total RNA and 0.5 μg of oligo (dT15) to 12.5 μL of sterile, distilled water and heated to 72 °C for 5 min. After cooling the samples on ice, 10 mM dithiothreitol (DTT), 0.5 mM of each dNTP, 5× first strand buffer and 200 U Superscript II RNase H–Reverse Transcriptase (Promega, Madison, WI, USA) was added. The mixture was stabilized at 25 °C for 10 min and subsequently incubated at 42 °C for 60 min for the RT reaction. Thereafter, the temperature was raised to 70 °C for 15 min to inactivate the reverse transcriptase. The synthesized cDNA was stored at −20 °C until use.

Quantitative real-time PCR

Quantitative real-time PCR was performed in a 20 μL PCR reaction containing a final concentration of 10 μL 2 × SYBR Premix DimerEraser and 0.4 μL ROX (passive reference dye) (Promega, Madison, WI, USA), 2 μL of cDNA of each sample, and 0.5 μM each of the forward and reverse primers. The thermal cycling profile consisted of four steps: (1) denaturation at 95 °C for 30 s; (2) amplification for 45 cycles of denaturation at 95 °C for 5 s, annealing at 55 °C for 30 s, and extension at 72 °C for 1 min; (3) melting curve by 95 °C for 15 s, 60 °C for 1 min, and 95 °C for 30 s; and (4) cooling at 4 °C. All reactions were conducted in triplicate using a 7500 Real-Time PCR System (Applied Biosystems; Foster City, CA, USA). A non-template nuclease-free water control was included for each run. The primers (Applied Biosystems; Foster City, CA, USA) for P-selectin, E-selectin, VEGF, TF, TLR4, IFN-α, β, γ, IL-1, 6, 8, and hypoxanthine phosphoribosyl-transferase (HPRT) are listed in Table 1. SyBgreen Gene Expression Assays (Promega, Madison, WI, USA) were used in our study. To evaluate the relative quantification of cDNA, the cycle threshold (Ct) value of the target genes for each sample were determined and normalized to the Ct-values of the housekeeping gene, HPRT. The results were presented as the fold change using the 2−ΔΔCt method.

Table 1 Sequences of the oligonucleotide primers used for real-time RT-PCR.

Gene specificity		Oligonucleotide sequences (5′–3′) of primers	
HPRTa	F	GTGATAGATCCATTCCTATGACTGTAGA	
	R	TGAGAGATCATCTCCACCAATTACTT	
P-selectinb	F	AACGGAGGGGAGGCAACAAGAC	
	R	GTGAGGGGACCAAGAGAAG	
E-selectinb	F	ATGATTGCTTCACAGTTTCTCT	
	R	TCACATGTCACAGCTTTACACG	
TLR-4c	F	AAGGTTATTGTCGTGGTGT	
	R	CTGCTGAGAAGGCGATAC	
VEGFd	F	ATGAACTTTCTGCTGTCTTGGGTG	
	R	TCACCGCCTCGGCTTGTCACATCT	
TFd	F	TTCAAGAC (A/C) ATT (T/C) TGGAGTGG	
	R	AGGGGGAGTTGGTAAAC	
IL-1d	F	ACTTCCTGGGGACGGCATGGATAAA	
	R	GCATCATTCAGGATGCACTGGTGGT	
IL-6d	F	GCTGCTTCTGGTGATGGCTACTGCC	
	R	TGAAACTCCACAAGACCGGTGGTGA	
IL-8d	F	AGCCCGTGTCAACATGACTTCC	
	R	GAATTGTGTTGGCATCTTTACTGA	
IFN-αd	F	ATGGCCCCAACCTCAGCCTTC	
	R	TCACTCCTTCTTCCTGAGTCT	
IFN-βd	F	ATGGCTAACAAGTGCATCCTCCAA	
	R	TCAGTTCCGGAGGTAATCTGTAAG	
Notes.

HPRT hypoxanthine phosphoribosyl-transferase

IFN-α interferon α

IL-1 interleukin 1

F forward primer

R reverse primer

a Zhu et al. (2008).

b Massaguer et al. (2003).

c Liu et al. (2009).

d Bensaude et al. (2004).

Protein quantification by ELISA

The concentration of IFN-α, IFN-γ, and TGF-β in the cell culture supernatants was assayed according to the manufacturer instructions of commercial ELISA kits (R&D Systems; Minneapolis, MN, USA).

Statistical analysis

SPSS software was used for all statistical evaluations (SPSS Version 16.0, SPSS Inc., Chicago, Illinois, IL, USA). A one-way analysis of variance (ANOVA) was used to test for differences between the groups, and the data were analyzed via a non-parametric Dunnett’s test. The results are presented as the means ± SEM. P-values less than 0.05 were considered to be statistically significant.

Results

TCID50 of CSFV

Puce cytoplasm reveals which ECs were infected with CSFV (Fig. 1). The results are expressed as 50% of the tissue culture infective dose (TCID50): 103 TCID50/mL is equivalent to 10 μg/mL of the CSFV C-strain.

Figure 1 ECs infected with CSFV titrated by indirect immunofluorescence.

The deep brown in the cells indicates which parts were infected with CSFV.

mRNA expression of E-selectin, P-selectin, VEGF, TF, TLR-4, IFN-α, IFN-β and IL-1, 6, and 8 on ECs following infection with CSFV in the presence or absence of APS

Compared with medium alone (Fig. 2A), the mRNA expression of E-selectin was significantly increased 3 h after stimulation with CSFV in Group 4 (P = 0.043) and was even higher in Group 6 (P < 0.001). After 6 h of stimulation, the mRNA expression of E-selectin decreased in Group 3 (P < 0.001) but increased in Group 5 (P < 0.010).

Figure 2 The relative mRNA expression of P-selectin (A), E-selectin (B), VEGF (C) and TF (D) in porcine ECs.

The relative mRNA expression in porcine ECs cultured with either medium alone (Medium), pretreated with APS (APS pre-treated), APS, CSFV, CSFV following pretreatment with APS (CSFV + APS pre-treated), or CSFV plus APS (CSFV + APS). Different texture filled in bars shows different culture conditions which illustrated in A. Gene expression was analyzed by SYBE Green RT-PCR. Data are presented as the means ± SEM of three independent experiments. Within the same time point: *, P < 0.05; **, P < 0.01; ***, P < 0.001.

After 3 h of stimulation with CSFV (Fig. 2B), a significant decrease in the mRNA expression of P-selectin (P = 0.001) compared to medium alone was detected in Groups 2 and 3, which were treated with APS. At 6 h post-stimulation, the mRNA expression of P-selectin was significantly increased in the groups stimulated with CSFV (Groups 4, 5, and 6) compared with groups that were not stimulated with CSFV (Groups 1, 2, and 3) (P < 0.010). The expression of P-selectin was lower after stimulation with CSFV plus pretreatment with APS compared with the group stimulated with CSFV alone (P < 0.001), whereas the expression was higher after stimulation with both CSFV and APS simultaneously (P < 0.001).

VEGF mRNA expression in the ECs (Fig. 2C) was higher in the presence of APS (P = 0.010) or CSFV (P = 0.025) alone but decreased in the presence of both CSFV and APS compared to the medium alone at 3 h post-stimulation. Moreover, the mRNA expression of VEGF was higher following stimulation with CSFV after pretreatment with APS compared to all the other test groups (P = 0.001). After 6 h of stimulation, VEGF mRNA expression was significantly increased in all stimulated groups compared to the medium alone (P < 0.001). VEGF mRNA expression was higher in the presence of CSFV plus APS (Groups 5 and 6) than in the presence of CSFV alone (P < 0.050). In addition, the expression of VEGF mRNA was higher in the presence of CSFV following pretreatment with APS than in the group simultaneously treated with CSFV and APS (P < 0.001).

After 3 h of stimulation, the expression of TF mRNA (Fig. 2D) was higher in the presence of APS alone compared to APS pre-treatment alone (P = 0.005); TF mRNA expression in the ECs was significantly increased in the presence of CSFV (P ≤ 0.010). After 6 h of stimulation, TF expression was higher following APS pre-treatment alone than in the presence of APS alone (P = 0.018). TF mRNA expression was increased in the presence of CSFV alone (P = 0.001) but not following CSFV plus APS.

TLR4 mRNA expression (Fig. 3A) in the ECs was significantly increased in the presence of CSFV (P < 0.010) but not in the presence of CSFV plus APS pre-treatment after 3 h of stimulation. However, TLR4 mRNA expression in the ECs was significantly increased in the presence of CSFV plus APS when administered simultaneously (Group 6) (P < 0.010), which was higher than in the presence of CSFV alone (P = 0.016). At 6 h post-stimulation, the mRNA expression of TLR4 was significantly increased in the presence of APS alone (P < 0.001) compared to the medium controls.

Figure 3 Relative mRNA expression of TLR4 (A), IFN-α (B), IFN-β (C), IL-1 (D), IL-6 (E) and IL-8 (F) in porcine ECs.

Relative mRNA expression in porcine ECs cultured with either medium alone (Medium), pretreatment with APS (APS pre-treated), APS, CSFV, CSFV following pretreatment with APS (CSFV + APS pre-treated), or CSFV plus APS (CSFV + APS). Different texture filled in bars shows different culture conditions which illustrated in A. Gene expression was analyzed by SYBE Green RT-PCR. Data are presented as the mean ± SEM of three independent experiments. Within the same time point: *, P < 0.05; **, P < 0.01; ***, P < 0.001.

At 3 h post-stimulation, the expression of IFN-α mRNA increased (Fig. 3B) in the presence of APS alone (P = 0.050) and CSFV alone (P = 0.006). At 6 h post-stimulation, the mRNA expression of IFN-α was increased in Groups 2 and 5 (P < 0.001). At 3 h after stimulation, IFN-β mRNA expression (Fig. 3C) was lower in the presence of APS (P = 0.026) or CSFV alone (P = 0.016) than in medium control group; however, it was higher in the presence of CSFV pretreated with APS (P = 0.050). At 6 h after stimulation, IFN-β mRNA was decreased in the majority of the groups except for Group 2 compared to the medium alone.

At 3 h after stimulation, the mRNA expression of IL-1 (3D) was increased in Group 3 (P = 0.012), 4 (P = 0.005), 5 (P = 0.001), and 6 (P = 0.009). At 6 h post-stimulation, IL-1 mRNA expression was only increased in the presence of APS alone (P < 0.001).

Compared to the medium control at 3 h after stimulation, IL-6 mRNA expression (Fig. 3E) was decreased in the ECs pretreated with APS (P = 0.026), while it increased in the presence of APS alone (P = 0.033). At 6 h after stimulation, IL-6 mRNA expression was increased in Groups 2 and 3, in which APS was added at different times (P < 0.050). IL-6 mRNA expression was significantly increased in the presence of CSFV pretreated with APS (P = 0.008).

At 3 h after stimulation, the expression of IL-8 mRNA (Fig. 3F) was significantly increased in Groups 3 and 4 (P = 0.002) and was increased in the group simultaneously treated with CSFV plus APS (Groups 5 and 6) (P = 0.001). At 6 h after stimulation, IL-8 mRNA expression was significantly increased in Groups 3 (P = 0.001) and 4 (P = 0.030). IL-8 mRNA expression was much lower in the presence of CSFV plus APS (Group 6) than in the presence of CSFV alone (P = 0.043).

Cytokine production following CSFV is modulated in ECs treated with APS

IFN-α expression in the EC supernatants was increased in all the stimulated groups (Groups 2–6; Fig. 4A); particularly following stimulation with APS (Groups 2 and 3). The expression of IFN-α was significantly increased after stimulation with CSFV but was lower in the presence of CSFV plus APS (Groups 5 and 6). Moreover, IFN-α expression was lower in the presence simultaneously of CSFV plus APS than in the presence of CSFV following pre-treatment with APS.

Figure 4 Enzyme-linked immunosorbent assay of IFN-α, IFN- γ and TGF-β in cell culture supernatants.

Porcine ECs were cultured with either medium alone (Medium), pretreated with APS (APS pre-treated), APS, CSFV, CSFV following pretreatment with APS (CSFV + APS pre-treated), or CSFV plus APS (CSFV + APS). Data are presented as the means ± SEM of three independent experiments. Comparisons between all groups P < 0.001.

IFN-γ expression in the EC supernatants was increased in the presence of APS alone but decreased significantly in the presence of CSFV (Groups 4, 5 and 6; Fig. 4B). In addition, IFN- γ expression was much lower following stimulation with CSFV plus APS (Groups 5 and 6) compared to the presence of CSFV alone and was the lowest in the presence of CSFV following pre-treatment with APS.

TGF-β expression (Fig. 4C) in the EC culture supernatants was increased in the presence of APS alone, CSFV alone, and CSFV plus APS, but decreased in the presence of CSFV following pretreatment with APS.

Discussion

CSFV infection induced a functional decrease in the pro-coagulant activity of the ECs, which were highly susceptible to CSFV infection (Campos et al., 2004). The inhibitory effects of APS on porcine viruses infection have verified (Xue et al., 2017) and APS has been observed to regulate the levels of immunomodulating compounds in many different cell types (Wu et al., 2017; Huang et al., 2013). The manifestation of adhesion molecules, including P-selectin and E-selectin, was inhibited by APS and persisted while the ECs were stimulated with CSFV, which could promote the expression of both P- and E-selectin (Figs. 2A and 2B). Similarly, the increase of TF (Fig. 2D), TGF-β (Fig. 4C), and IL-8 (Fig. 3F) production in response to CSFV was inhibited by APS.

The coordinated expression of adhesion molecules leads to the localization of leukocytes to a tissue or organ, and members of the selectin family initially mediate the transition from rapidly flowing the bloodstream to rolling along the endothelium (Lawson et al., 2000). E-selectin is an early mediator of leukocyte-endothelial adhesion and is expressed on activated endothelium (Whalen et al., 1999). The data presented in the present study unequivocally demonstrated that APS inhibited the activation of E-selectin caused by CSFV which was also delayed by APS pre-treatment, indicating that APS has a preventive effect on the hemorrhaging stimulated by CSFV. However, the addition of APS following infection with CSFV induced the co-stimulation of E-selectin which fade away after 3 h, indicating that the addition of APS may enhance vaccine effectiveness, but can’t maintain for a long time (Fig. 2A).

Our results also revealed that VEGF expression was increased by CSFV and could be further enhanced with the pre-treatment of APS (Fig. 2C). This finding indicates that CSFV and APS may elicit an angiogenic response via VEGF (Pasyk & Jakobczak, 2004; Ferrara & Keyt, 1997). Furthermore, the release of TF may be important in mediating the activation of both coagulation and fibrinolytic mechanisms (Levi et al., 1994). We found that APS inhibited the expression of TF, which was upregulated by CSFV. The inhibition of APS appeared later than upregulation of CSFV (Fig. 2D). This phenomenon involved in disseminated intravascular coagulation and thrombocytopenia during the progression of CSFV (Penrith, Vosloo & Mather, 2011).

As a pattern recognition receptor, TLR4 can activate the innate immune response to both bacterial and viral pathogens, including RNA viruses (Rassa et al., 2002). We demonstrated that the immune response mediated by TLR4 in ECs was induced by CSFV, which contributes to the ability of CSFV to replicate without exerting a cytopathic effect (CPE) (Bensaude et al., 2004; Jiang et al., 2008). We found that APS can inhibit the expression of TLR4 in ECs following infection with CSFV (Fig. 3A) at a later time, while co-stimulate and promote it at an early time point, indicating that APS can regulate innate immunity and vessel remodeling via TLR4 expression which has been proved in macrophages lately (Zhou et al., 2017).

The suppression of IFN-β plays a role in the inability of porcine viruses to spread and cause disease in susceptible animals (Chinsangaram, Piccone & Grubman, 1999); however, this cytokine was upregulated when the ECs were pretreated with APS (Fig. 3C). Our results demonstrate that CSFV inhibited the expression of IFN- γ (which always fails to be produced in cells infected with CSFV (Bensaude et al., 2004)). IFN-γ profoundly inhibited African swine fever virus replication in porcine macrophages (Esparza, Gonzalez & Vinuela, 1988), as well as the cytopathic effect produced by CSFV (Xia et al., 2005); however, this inhibition remained unchanged following the addition of APS (Fig. 4B).

IL-1, IL-6, and IL-8 are typical examples of the multifunctional cytokines involved in the regulation of the immune response, hematopoiesis, and inflammation (Molbak, Lisse & Aaby, 1996; VanCott et al., 2000). Following an infection with CSFV, there is an initial and brief increase in the transcript levels of IL-1 and 8 (Bensaude et al., 2004), which was confirmed in our study (Figs. 3D–3F); the increase in IL-8 was inhibited by APS.

In conclusion, the data from the present study indicate that APS induced the production of immune factors and prevented an increase in adhesion molecules, which were inhibited and activated by CSFV, respectively. In some cases, the addition of APS induced the co-stimulation of factors associated with CSFV, indicating that the addition of APS may enhance vaccine efficacy. Thus, APS may have an modulation effect on ECs exposed to CSFV and utilize different mechanisms to influence the adhesion-related immune response of ECs during the hemorrhagic treatment. However, in vitro data is not sufficient to conclude that APS has a preventive effect on the hemorrhaging induced by CSFV. These possibilities warrant further study to evaluate whether APS may be an effective adjuvant in in vivo clinical vaccine trials against CSFV.

Supplemental Information

Data S1 Raw data of relative mRNA expression of IL-1, IL-6 and IL-8 in porcine ECs

Click here for additional data file.

Data S2 Raw data of the relative mRNA expression of P-selectin in porcine ECs

Click here for additional data file.

Data S3 Raw data of the relative mRNA expression of IFN-β in porcine ECs

Click here for additional data file.

Data S4 Raw data of the relative mRNA expression of E-selectin in porcine ECs

Click here for additional data file.

Data S5 Raw data of the relative mRNA expression of TLR-4 in porcine ECs

Click here for additional data file.

Data S6 Raw data of the relative mRNA expression of VEGF, TF and IFN-α in porcine ECs

Click here for additional data file.

Additional Information and Declarations

Competing Interests

Author Contributions

Data Availability

The authors declare there are no competing interests.

Zengyu Zhuge conceived and designed the experiments, performed the experiments, analyzed the data, wrote the paper, prepared figures and/or tables.

Yanpeng Dong reviewed drafts of the paper.

Liuan Li and Tianming Jin contributed reagents/materials/analysis tools.

The following information was supplied regarding data availability:

The RT-PCR raw data have been submitted as a Supplemental File.

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
