# Peer review of "Effects of astragalus polysaccharide on the adhesion-related immune response of endothelial cells stimulated with CSFV in vitro"

_PeerJ, doi:10.7717/peerj.3862_

## Round 0.1 · original submission · Major Revisions

The paper should be carefully revised following the reviewers comments, in particular those of reviewers 2 and 3.

Reviewer 1 ·

Basic reporting

It is a solid, well-written article on the effects of Astragalus polysaccharide on porcine endothelial cells. Figures are good, literature adequate.

Experimental design

The article and data are original enough to be published in the journal. The experimental design is well prepared, experiments are good and results are sound. The only problem is the lack of definition of the Astralagus polysaccharide. As it is crucial for the investigation, the respective information has to be added.

Validity of the findings

Data are find, sound and interesting. Conclusion is fully based on results. List of references is adequate and relevant.

Additional comments

An overall solid paper with potential clinical and/or industrial interests has one problem - the lack of definition of the Astralagus polysaccharide. As it is crucial for the investigation, the respective information has to be added. It is not possible to publish biological study without attempting the fully describe the material used in the study.

Reviewer 2 ·

Basic reporting

1. The introductions need more details about the relationships between ECs cells, hemorrhagic diseases, and the factors tested in this studies.
2. In the Discussion section, please refer to related figures in text. The authors did that in the Results section, but they are also needed in the Discussion section.

Experimental design

1. The authors need to indicate the catalog numbers or concentrations of the antibodies used in this study.
2. In Line 84, did the authors add serum and antibiotics to the medium?
3. In Lines 93-95, please write the amounts of HPR-labeled ABC and DAB were added into each well.
4. In Line 240-243, it looked like P-selection expression occurred later than E-selection expression in Figures 2A and 2B. Since the WPBs were released quickly, the authors cannot observe that kind of difference through qRT-PCR, which is used to study the changes at transcriptional level. The authors need to explain more clearly about their proposed mechanisms.
5. In Lines 366-367, what does “puce staining” mean?
6. In Lines 369-370, the authors need to refer the different bars to the culture conditions. Also do this in figure legends 3 and 4. Also, add “either” before “medium alone”, and replace “and CSFV plus APS” by “or CSFV plus APS”.

Validity of the findings

1. My only major concern is the claims about APS’s functions. In this work, the authors studied the gene expression profiles related to cell adhesion and immune response, but did not do functional assays to confirm APS’s protective role during CSFV infection nor use APS to treat hemorrhagic infection.
2. In the abstract, the authors concluded that APS is a potential adjuvant for CSFV vaccine. However, in the last paragraph of Introduction, the authors claimed that the adjuvant part was studied in their previous work, while the aim of this study was to identify a method for hemorrhagic treatment. Please make the conclusion clear and consistent throughout this manuscript.

Additional comments

The paper studied the immunomodulatory effects of Astragalus polysaccharide, the key active component extracted from a traditional Chinese medicinal herb, on endothelial cells’s immune response to classical swine fever virus. The authors stimulated the ECs with CSFV in the presence of APS and investigated the gene expression profiles of key factors related to cell adhesion, cell growth and immune response. They observed that the addition of APS could downregulate the expression of selectins, tissue factor, TLR-4, interferon and TGF beta, thus indicating APS’s potential as an adjuvant for CSFV vaccine.

The authors used qPT-PCR and ELISA to quantify gene expression, which is commonly used method in this area; the statistical analysis part is good.

1. In Lines 369-370, the authors need to refer the different bars to the culture conditions. Also do this in figure legends 3 and 4. Also, add “either” before “medium alone”, and replace “and CSFV plus APS” by “or CSFV plus APS”.
2. In Line 371, RT-PCR is real-time PCR here, the “real-time” can be deleted.
3. In Lines 371 and 377, it is “SYBR Green” not “SYBgreen”.

Reviewer 3 ·

Basic reporting

In the study entitled ‘Effects of astragalus polysaccharide on the adhesion-related
Immune response of endothelial cells stimulated with CSFV in vitro’, the authors have studied the effects of Astragalus polysaccharide on porcine endothelial cells. They have quantified the mRNA expression levels of E and P-selectins ,TNF, VGF and cytokines after CSFV infection and the effect of APS either by pretreatment or addition along with CSFV. The have also studied the levels of cytokines in these cultures by ELISA. Overall, the experiments are well formulated but the manuscript lacks a detailed discussion of the results and interpretation of differences observed at different time points after stimulation. The authors should discuss their observations in the light of what has already been published in the literature in case of CSFV and other viral infection. APS has been observed to regulate the levels of immunomodulating compounds in many different cell types before (Zhou L, Sci Rep, 2017; Xue H, Sci Rep, 2017; Wu S, Sci Rep, 207 etc), whether these effects on Endothelial cells are specific for CSFV or true for most viral infection should be stated. Also, APS has been shown in previous studies to have an effect on many different cell types including myocardial functions, neuronal cells, macrophages etc, therefore, in general, its effects on various different cell types have been studied and its use as an adjuvant in vaccines against viral infections may be generic and not specific for CSFV and endothelial cells.

Experimental design

Line 42-46) -The basis of the study that author’s state in introduction is that vascular endothelial cells play a role in recruitment of immune cells and influence the outcome of immune response, but not sufficient literature is cited to support this.
Many different cell types are involved in innate immune defense in viral infections, and the authors have cited their own study on the effect of Astralagus polyscachharide on CSFV infection in peripheral blood mononuclear cells, published in Plos One in 2012. But they fail to convince in the introduction and discussion that ECs are infact important cells involved in primary line of immune defense against CSFV infection. Other such as Borca M, 2008 (Virus Res), Dong X, 2013 (Virology J), Glaude , 2008 (J virol) have studied CSFV infection in different cell types and should be cited.
(Line 51-70)- The background on Astragalus polysaccharide is hugely similar to the previous study the authors have published on the role of APS in PBMCs (Zhuge ZY, 2012, Plos One) and should be re written.

Validity of the findings

No comments

---

## Round 0.2 · Minor Revisions

Please, consider the revision requested by the reviewer 1. Try to give more information on the Astralagus polysaccharide. Check line 73!

Reviewer 1 ·

Basic reporting

Generally, my review of the new version remains the same: An overall solid paper with potential clinical and/or industrial interests has one problem - the lack of definition of the Astralagus polysaccharide. As it is crucial for the investigation, the respective information has to be added. It is not possible to publish biological study without attempting the fully describe the material used in the study.

The authors revised or explained all requests from the second reviewer., however, in case of my request, nothing really changed. Line 73 does not explain anything. Additional references support the claims, we there is still a need to explain the material, i.e., composition, purity etc.

Experimental design

No comments

Validity of the findings

No comments

Additional comments

From the reviewer;s pioint of view, nothing really changed. Line 73 does not explain anything. Additional references support the claims, we there is still a need to explain the material, i.e., composition, purity etc.

Reviewer 2 ·

Basic reporting

No comment

Experimental design

No comment

Validity of the findings

No comment

Additional comments

The revised version addresses all my concerns and it is well-written. I recommend to accept the manuscript for the publication on PeerJ.

---

## Round 0.3 · accepted · Accept

This is an overall solid paper with potential clinical and/or industrial interests.